# Interdiffusion and Intermetallic Compounds at Al/Cu Interfaces in Al-50vol.%Cu Composite Prepared by Solid-State Sintering

**DOI:** 10.3390/ma14154307

**Published:** 2021-07-31

**Authors:** Dasom Kim, Kyungju Kim, Hansang Kwon

**Affiliations:** 1Department of Materials System Engineering, Pukyong National University, Busan 48547, Korea; dasom.kim@f.mbox.nagoya-u.ac.jp; 2The Industrial Science Technology Research Center, Pukyong National University, Busan 48547, Korea; ngm13@ngm.re.kr; 3Department of R&D, Next Generation Materials Co., Ltd., Busan 48547, Korea

**Keywords:** aluminium, copper, metal matrix composite, solid-state sintering, interdiffusion, intermetallic compound, thermal conductivity

## Abstract

Al–Cu composites have attracted significant interest recently owing to their lightweight nature and remarkable thermal properties. Understanding the interdiffusion mechanism at the numerous Al/Cu interfaces is crucial to obtain Al–Cu composites with high thermal conductivities. The present study systematically investigates the interdiffusion mechanism at Al/Cu interfaces in relation to the process temperature. Al-50vol.%Cu composite powder, where Cu particles were encapsulated in a matrix of irregular Al particles, was prepared and then sintered at various temperatures from 340 to 500 °C. Intermetallic compounds (ICs) such as CuAl_2_ and Cu_9_Al_4_ were formed at the Al/Cu interfaces during sintering. Microstructural analysis showed that the thickness of the interdiffusion layer, which comprised the CuAl_2_ and Cu_9_Al_4_ ICs, drastically increased above 400 °C. The Vickers hardness of the Al-50vol.%Cu composite sintered at 380 °C was 79 HV, which was 1.5 times that of the value estimated by the rule of mixtures. A high thermal conductivity of 150 W∙m^−1^∙K^−1^ was simultaneously obtained. This result suggests that the Al-50vol.%Cu composite material with large number of Al/Cu interfaces, as well as good mechanical strength and heat conductance, can be prepared by solid-state sintering at a low temperature.

## 1. Introduction

In recent times, lightweight materials with excellent heat dissipation characteristics have found application in various industries: electronics (displays and semiconductors), thermal management (heat sinks), and transportation (hybrid and electric automobiles). Metal matrix composites (MMCs) have attracted attention for several decades as promising materials that can realise superior mechanical [1,2,3], electrical [4,5,6] and thermal properties [7,8,9,10]. MMCs can also overcome the limitations of pure metals and metallic alloys. The key factors to realising high-performance MMCs are achieving a uniform mixing state and controlling the interface between dissimilar materials. Powder metallurgy is regarded as an effective method to prepare MMC with a uniform mixing state [11,12]. In MMCs prepared by powder metallurgy, large number of interfaces is generated between dissimilar materials, which should be controlled. The thermal mismatches at the interfaces cause stress gradients and generate cracks, which could affect the interfacial mechanical and thermal properties. Additionally, ensuring heat transfer across the interfaces is critical to achieve superior thermal properties of MMCs [13].

There is growing interest in Al–Cu composites that are lightweight and simultaneously offer remarkable electrical and thermal properties [14,15,16]. At even lower temperatures than the melting point of Al, interdiffusion in the solid state leads to the formation of Al–Cu intermetallic compounds (ICs) at the Al/Cu interfaces. When ICs are formed, chemical reactions can strengthen the Al–Cu bonds, thereby enhancing the mechanical strength and thermal stability. However, the non-metallic covalent bonds in the ICs prevent free-electron conduction (which is crucial for thermal conductivity) and thus, significantly degrade the electrical and thermal conductivities of the Al–Cu MMCs. Therefore, understanding the interdiffusion mechanism and controlling the degree of IC formation at the Al/Cu interface are critical for good thermal conductivity. Školáková et al. reported that the mechanical strength and hardness are influenced by the presence of the CuAl_2_ phase [17]. The phases are very stable and when this phase is contained in the aluminum alloys could be precipitated during heat treatment and improve thermal properties [18,19].

The interdiffusion and IC formation mechanisms at the Al/Cu interfaces have been studied over a wide range of temperatures in Al–Cu composites produced by processes such as welding [20,21,22], casting [23], hot pressing [24] and rolling [25]. The Al–Cu phase diagram predicts the existence of five equilibrium phases: Cu_9_Al_4_, Cu_3_Al_2_, Cu_4_Al_3_, CuAl, and CuAl_2_. However, the compositions of the ICs, which are actually formed during manufacturing, depend on the process temperature and time [26,27,28,29,30,31]. For short diffusion times, CuAl_2_ and Cu_9_Al_4_ are generally formed, while for longer diffusion times at high temperature, CuAl and Cu_3_Al_2_ are also formed. Many researchers have interpreted the IC formation mechanism and sequence with respect to the formation energy and diffusivity of the IC phases. CuAl_2_ with the lowest formation energy and Cu_9_Al_4_ with high atomic diffusivity are initially formed at the Al/Cu interface. Other phases such as Cu_3_Al_2_, Cu_4_Al_3_, and CuAl, which have low diffusivity, are formed later [32,33]. However, reports are scarce on the formation and growth behaviour of ICs in the low-temperature range. Furthermore, the effect of ICs on the thermal properties of Al–Cu composites prepared by powder metallurgy has not been studied systematically.

We previously prepared Al–Cu composites with a high content ratio of Cu by powder metallurgy at 520 °C and studied the effect of ICs on the mechanical and thermal properties. The ICs formed at the Al/Cu interfaces enhanced the hardness of the composites but had a negative effect on the thermal conductivity due to large number of Al/Cu interfaces [30]. The present study is a further step towards investigating the effect of Al/Cu interfaces on the mechanical and thermal properties of Al–Cu composites. We systemically investigated the effect of the process temperature on interdiffusion at the Al/Cu interface (i.e., the formation and growth mechanisms of Al–Cu ICs). We prepared Al–Cu composites by solid-state sintering over a temperature range of 340 to 500 °C. The microscale observation and compositional analysis of the Al/Cu interfaces during interdiffusion, respectively, specifically for the temperature range of 360 to 400 °C. The interdiffusion layer at the Al/Cu interfaces and its temperature dependence with respect to the formation energy and the growth rate of ICs were investigated. Vickers hardness values were measured to determine the mechanical stability of the Al/Cu interface and the mechanical strength of the Al–Cu composite. Finally, the thermal conductivity was evaluated, and the results were discussed with reference to the equations relating the IC thickness to the thermal conductivity.

## 2. Materials and Methods

### 2.1. Preparation and Analysis of Al–Cu Powder

Al and Cu powders with a purity of 99.9% and particle size of 45 μm (Metalplayer) were used as raw materials. The Al and Cu powders were mixed in a volume ratio of 1:1 by ball milling at a rotational speed of 300 rpm for 24 h (SMBL-6, SciLab Mix™, Programmable Ball Mill, Seoul, Korea). The weight of each Al and Cu powder at a 1:1 volume fraction was calculated using the theoretical density of Al (2.70 g∙cm^−3^) and Cu (8.96 g∙cm^−3^). The 23.16 g of Al powder and 76.84 g of Cu powder were poured into container in order to prepare 100.00 g of Al–Cu powder. For the ball milling, a stainless-steel container and ZrO_2_ and stainless-steel balls with diameters of 15 and 3.2 mm were used, respectively. The powder-to-ball volume ratio in the container was 1:3. Heptane (20 mL) was used as the process control agent to prevent cold welding of metal particles during ball milling. The morphology of the ball-milled powder was observed by field emission scanning electron microscopy (FE-SEM, MIRA 3 LMH In-Beam, TESCAN, Brno, Czech Republic). To determine the mixing state of the ball-milled powder particles, the cross-section of the ball-milled powder was observed by field-emission electron probe micro-analysis (FE-EPMA, JXA-8530F, JEOL Ltd., Tokyo, Japan). The phases in the ball-milled powder were detected using X-ray diffraction (XRD, Ultima IV, Rigaku, Tokyo, Japan) with a Cu Kα radiation source (λ = 1.5148 Å, 40 kV, and 40 mA) in the 2θ scanning range of 20–80°.

### 2.2. Preparation and Characterization of Al–Cu Composite

The ball-milled powder was sintered at 340, 360, 380, 400, 420, 440, 460, 480, and 500 °C under a compression pressure of 50 MPa for 5 min (SPS-321Lx, Fuji Electronic Industrial Co., Ltd., Tsurugashima, Japan) under vacuum (<0.8 Pa). The heating rate was 30 °C⋅min^–1^. The Al-50vol%Cu composites sintered at x °C are denoted as ‘Al–Cu(x °C)’. The density of the composite material was measured using the Archimedes method. The relative density was calculated from the volume fractions of Al and Cu using the rule of mixtures. The phases in the Al–Cu composites were detected via XRD in the 2θ scanning range of 20–50°. The microstructures of the sintered Al–Cu composites were analysed using FE-SEM and energy dispersive X-ray spectroscopy (EDS, EX-400, HORIBA, Kyoto, Japan). EDS line scans were performed across the Al/Cu interfaces to analyse the ICs. The theoretical density was calculated using the rule of mixtures. The Vickers hardness (HM-101, Mitutoyo Corp., Kawasaki, Japan) was measured with a load of 0.3 kg for 15 s according to the JIS B 7725 and ISO 6507-2 standards. To determine the thermal conductivity (κ) of the Al–Cu composites, the heat diffusivity (α) and heat capacity ($C_{p})$ were measured at room temperature (20 ± 2 °C) using a laser flash device (LFA 467, Netzsch, Selb, Germany) in accordance with the ISO 22007-4, ISO 18755, and ASTM E1461 standards. The accuracy of the measuring device was ± 3% for the heat diffusion coefficient and ± 5% for the heat capacity. κ was then calculated using Equation (1).
(1)$$\kappa =\rho \times \alpha \times C_{p}$$
where ρ is the density of the material, α is the thermal diffusivity, and $C_{p}$ is the heat capacity.

## 3. Results and Discussion

### 3.1. Formation of Intermetallic Compounds at Al/Cu Interface in Al–Cu Composite

Figure 1 provides information on the Al–Cu powder prepared by ball milling. Figure 1a shows the FE-SEM images of the ball-milled Al–Cu powder. The particle size of the ball-milled Al–Cu powder exceeded 100 μm, which is greater than that of the raw Al and Cu powders (45 µm). This might have resulted from the adhesion of the 45 µm raw Al powder particles to the 45 µm raw Cu powder particles during ball milling due to the mechanical energy. When a ball-milled Al–Cu powder particle was analysed at high magnification, as shown in Figure 1b, small particles with a particle size of 40–50 μm were observed to be attached together to form a large particle. The small particles had either a spherical (marked as ‘A’) or an irregular (marked as ‘B’) shape. Particles A and B might be Cu and Al, respectively. To analyse the composition of the ball-milled particles in the mixing state, the cross-section of the ball-milled Al–Cu powder was observed by FE-EPMA. As shown in Figure 1c, the light grey and white regions are Al and Cu particles, which can be confirmed by the elemental mapping results for Al and Cu that were obtained (Figure 1d,e, respectively). Severely deformed Al particles surrounded the spherical Cu particles. It appears that the Cu particles were dispersed uniformly within the Al matrix. Each particle of the ball-milled Al–Cu powder was a form of “composite powder”, with the deformed Al particles encapsulating the spherical Cu particles. As a result of XRD (Figure 1f), only Al and Cu peak were detected. It indicates that the Al–Cu composite powder could be prepared solely by mechanical mixing without any chemical reaction. The prepared Al–Cu composite powder by ball milling in this study is suitable for preparing Al–Cu composites with a uniform mixing state and for investigating the thermal reaction during sintering at the Al/Cu interface.

The Al–Cu composite powder was sintered at various temperatures between 340 and 500 °C under a compression pressure of 50 MPa for 5 min. Figure 2a shows the dependence of the relative density of the Al-50vol%Cu composites on the sintering temperature in the range of 340 to 500 °C. The relative density of the Al–Cu composite increases with the sintering temperature, indicating that higher sintering temperatures accelerate the densification of the Al–Cu composite powder. The densification process was divided into three steps based on the densification rate: step 1 (340 to 400 °C), step 2 (400–460 °C), and step 3 (460–500 °C). The relative density increased linearly in step 1, and finally reached 100% in step 2, where the densification rate decreased. In steps 1 and 2, the reduction in porosity may be related to densification. In step 3, the relative density was over 100%, indicating the generation of a new phase with a density higher than that of Al.

To analyse the phase transformation, XRD analysis was performed on the Al–Cu composites sintered at 340, 380, 420, 460, and 500 °C in a wide 2θ scanning range of 35–55°; the peak profiles are shown in Figure 2b. The Al–Cu ICs such as CuAl_2_ and Cu_9_Al_4_ were formed, and their phase peaks were detected in all Al–Cu composites, along with the peaks of pure Al and pure Cu phases. This indicates that the reaction between Al and Cu occurred at the Al/Cu interfaces even at a lower sintering temperature of 340 °C. The densities of CuAl_2_ (4.42 g·cm^−3^) and Cu_9_Al_4_ (6.85 g·cm^−3^) are higher than that of Al. Al is mainly consumed to form such ICs. When the intensities of the peaks of IC phases were compared to those of Al and Cu, it was inferred that the degree of formation of ICs differs depending on the sintering temperature.

The XRD profiles in the narrow 2θ scanning range of 38–45° are shown in Figure 2c. Al, Cu, Cu_9_Al_4_, and CuAl_2_ phase peaks were observed, in decreasing order of their intensity. With an increase in the sintering temperature, the intensities of the Al (111) peak at ~38° and Cu (111) peak at ~43° decreased, whereas the intensities of the CuAl_2_ (110) peak at ~21° and Cu_9_Al_4_ (330) peak at ~44 ° increased. This indicates that an increase in the sintering temperature increases the consumption of Al and Cu towards the formation of Al–Cu ICs. The intensities of the CuAl_2_ and Cu_9_Al_4_ peaks, which were low in the Al–Cu composites sintered at 340 and 380 °C, drastically increased in the Al–Cu composites sintered at 420 °C; correspondingly, the intensities of the Al and Cu peaks decreased significantly. In the low-temperature range below 420 °C, the degree of formation of the Al–Cu ICs might be controlled. Among the Al–Cu ICs, the formation energy of the CuAl_2_ phase is lowest, then first formed at the Al/Cu interface and grow toward Al. The growth rate of Cu_9_Al_4_ highest in the temperature range of ~500 °C, then Cu_9_Al_4_ is formed at the CuAl_2_/Cu interface and grow toward CuAl_2_ [26,33]. Therefore, the Cu_9_Al_4_ layer fraction might increase when diffusion is hold at high temperature range. Therefore, Al–Cu composite sintered at 480 and 500 °C had relative density over 100% because large amounts of ICs were formed at Al/Cu interfaces.

Figure 3 shows the FE-SEM images and EDS mapping results for the elements Al and Cu across the cross-section of the Al–Cu composites sintered at 360 °C (Figure 3a,d), 380 °C (Figure 3b,e), and 400 °C (Figure 3c,f). The bright and dark regions coincided with the Cu and Al regions, respectively, according to the EDS mapping results. Spherical Cu particles were dispersed within the Al matrix. The content of Al and Cu in Al–Cu composites were analysed by EDS mapping result. As a result of EDS mapping, the volume fractions of Al:Cu in Al–Cu composite sintered at 360, 380 and 400 °C were 56:44, 50:50 and 52:48, respectively (Appendix A). The small error range within 5% indicates that the Al-50vol.%Cu composites were successfully prepared in this study. Some pores were observed at the interface between Al and Cu regardless of the sintering temperature. Except for the regions where the pores resided, Al and Cu were in contact with each other, forming interfaces. Thus, these Al–Cu composites can be used to investigate the interdiffusion mechanism at Al/Cu interfaces.

The Al/Cu interfaces in the Al–Cu composites sintered at 360, 380, and 400 °C were analysed using FE-SEM and EDS line scans, and the results are shown in Figure 4. The bright and the dark regions in the FE-SEM images in Figure 4a–c are the Cu and Al sides, respectively. EDS line scanning was conducted from the Cu side to the Al side through the Al/Cu interface without pores and cracks. As shown in Figure 4d–f, the EDS scan area can be divided into five regions from I to V based on the concentrations of Al and Cu. In regions I and V, only Cu and Al were counted in the line scan, indicating that regions I and V represent the Cu side and Al side respectively. Regions II, III, and IV between the Cu side (region I) and Al side (region V) collectively constitute the interdiffusion layer. Region II is a Cu-rich layer, whereas regions III and IV are Al-rich layers; however, the Al and Cu concentration gradients vary differently (gentle slope in Region III, and steep slope in Region IV). This indicates that the diffusion was stabilised only in region II with a gentle slope. Regions II and IV are in situ diffusion layers. The diffusivity of Al in Cu is much higher than that of Cu in Al. Al diffuses into the Cu side, forming vacancies in the Al side. The Cu atoms with lower diffusivity than Al diffuse into the Al side and occupy the vacancies. This is an indication that region III might be the precipitated layer of CuAl_2_. Moreover, regions II and IV might comprise Cu-rich solid solution + Cu_9_Al_4_ and Al-rich solid solution + CuAl_2_, respectively. The thickness of the stabilised CuAl_2_ layer gradually increases with an increase in the sintering temperature because of the high diffusivity. However, the ratio of the thickness of region II to that of region IV in the total interdiffusion layer is a variable.

The total thickness of the interdiffusion layer in Al–Cu sintered at 360 °C (6.1 μm) was similar to that in Al–Cu sintered at 380 °C (6.7 μm). However, the thickness increased twofold when the sintering temperature was increased to 400 °C (10 μm), indicating a higher increase in the interdiffusion rate. The thickness of the IC phase depends on the growth constant of IC. The thickness of the IC layer can be expressed by Equation (2) [22,34,35].
(2)$$W=kt^{n}$$
where *W* is the thickness of the IC layer, *k* is the growth constant, *t* is the diffusion time, and n is the time exponent. The rate constant *k* is exponentially proportional to the temperature [33].
(3)$${\mathrm{CuAl}}_{2}:\;k^{2}=9.1\cdot 10^{-3}\times \exp\left(-\frac{29300}{RT}\right)$$
(4)$${\mathrm{Cu}}_{9}{\mathrm{Al}}_{4}:\;k^{2}=3.2\cdot 10^{-2}\times \exp\left(-\frac{31600}{RT}\right)$$

In Equations (3) and (4) R is the gas constant (8.13 J∙mol^−1^∙K^−1^), and *T* is temperature. The growth rate of the IC phase would increase drastically at an elevated sintering temperature of 400 °C. Thus, the growth of ICs can be suppressed at temperatures below 400 °C. Let us consider the thickness fractions of regions II, III, and IV (see Table 1). The fraction of region IV, which might be the Al + CuAl_2_ layer, gradually decreased with an increase in the sintering temperature, which may be attributed to the instantaneous precipitation of CuAl_2_ from the Al-rich solid solution, i.e., the fast growth of CuAl_2_ towards the Al side. In Al–Cu (380 °C), the fraction of region III, which is likely to be the CuAl_2_ layer, was lowest, and the fraction of region II, which could be the Cu + Cu_9_Al_4_ layer, was the highest. This implies that the growth rate of CuAl_2_ is the highest in Al–Cu (380 °C). The growth constant k of the IC layer can be considered as the thermal diffusivity. According to Zhang et al., the diffusivity of CuAl_2_ (1.8 × 10^−13^ K∙m^−2^∙s^−1^) is lower than that of Cu_9_Al_4_ (1.4 × 10^−13^ K∙m^−2^∙s^−1^) at 300 °C. However, the diffusivities of CuAl_2_ and Cu_9_Al_4_ are the same at 400 °C (4.1 × 10^−13^ m^2^∙s^−1^) [27]. This implies that k(Cu_9_Al_4_) increases at a rate higher than k(CuAl_2_) in the temperature range of 300 to 400 °C. First, CuAl_2_ is formed at the Al/Cu interface towards Al and then, Cu_9_Al_4_ is formed towards CuAl_2_ at Cu/ CuAl_2_. As the sintering temperature increased to 400 °C, Cu_9_Al_4_ grew at an increased rate towards CuAl_2_. In this study, the diffusivity was not measured, but it is expected that the growth rate of Cu_9_Al_4_ would exceed that of Cu_9_Al_4_ in the range of 380 to 400 °C.

In order to analyse the average interdiffusion layer growth, we used the ImageJ program to measure area fraction of each phase such as Cu, Al, CuAl_2_ and Cu_9_Al_4_ shown in Figure 5.

Figure 5a–c shows the area fraction of Cu, Al, CuAl_2_ and Cu_9_Al_4_ in sintered Al–Cu (360 °C), Al–Cu (380 °C) and Al–Cu (400 °C), respectively. As a result of area fraction analysis shown in Figure 5. The area fraction of Al decreased, but Cu increased with increase of sintering temperature, indicating that the Al was more consumed than Cu. The ratio of area fraction of CuAl_2_ and Cu_9_Al_4_ coincided with thickness ratio analysed by EDS.

### 3.2. Effect of Intermetallic Compounds on Properties of Al–Cu Composite

The properties of composites with many interfaces between dissimilar materials depend significantly on the interface state. As ICs are formed at the interfaces, dissimilar materials can be strongly bonded, the movement of dislocations can be prevented, and consequently, the strength can be enhanced. However, electron scattering at the interfaces leads to a deterioration in the thermal and electrical conductivities. We evaluated the hardness and thermal conductivity, and these results were shown in Table 2 with relative density.

Figure 6 shows the Vickers hardness test results of the Al–Cu composites. The Vickers hardness values of Al–Cu (360 °C), Al–Cu (380 °C), and Al–Cu (400 °C) were measured to be 65.0 ± 8.6, 78.5 ± 15.6, and 69.8 ± 2.4 HV, respectively, which were higher than the expected Vickers hardness of Al-50vol.%Cu by the rule of mixtures (53 HV, marked with a dot line). This could be attributed to the much higher hardness values of Al–Cu ICs than those of pure Al and Cu.

We deem three factors important for the hardness, namely, the density, area fraction of each material, and crystallite size. Taking the relative density into consideration, the Vickers hardness of Al–Cu (360 °C), which has many pores shown indentation optical image, could be increased by reducing the porosity. Al–Cu (380 °C) had the highest Vickers hardness. Although the relative density of Al–Cu (400 °C) is higher than that of Al–Cu (380 °C), the Vickers hardness decreased. This necessitates the consideration of other factors affecting the Vickers hardness. With respect to the degree of IC formation, the hardness values of CuAl_2_ and Cu_9_Al_4_ were reported to be 324 HV and 549 HV, respectively [31,36], which are much higher than those of Al and Cu. In addition, more Al is consumed than Cu during the formation of ICs. Therefore, we expected that Al–Cu (400 °C), where larger amounts of ICs were formed, would show a higher Vickers hardness.

However, our results were not in accordance with our expectation. Finally, with respect to the crystallite sizes of Al and Cu by using the Scherrer equation and XRD profiles. While the crystallite size of Al was constant, the crystallite size of Cu increased with the sintering temperature. The crystallite sizes of Cu in Al–Cu (340 °C), Al–Cu (380 °C), and Al–Cu (420 °C) were 38, 49, and 73 nm, respectively. This indicates that increase of crystallite size in Cu was accelerated between 380 and 420 °C. We expect that the Vickers hardness of Al–Cu (400 °C) would be lower than that of Al–Cu (380 °C) due to increased crystallite size. This could be corroborated by carrying out further analysis, such as transmission electron microscopy (TEM), in the future. From the results of the Vickers hardness tests, it is concluded that a sintering temperature of 380 °C may be suitable for preparing a strengthened Al–Cu composite with a fine microstructure.

Figure 7 shows the variations in the thermal conductivity and relative density of the Al–Cu composites with the sintering temperature in the range 340 to 500 °C. The thermal conductivity as well as the relative density increased with sintering temperature (Figure 2a) in the range 340 to 420 °C, and reached to 155 W∙m^−1^∙K^−1^. Although the relative density gradually increased with the increase in sintering temperature, the thermal conductivity decreased beyond a sintering temperature of 420 °C. The thermal conductivity might have deteriorated due to the growth of ICs at the Al/Cu interfaces.

The thermal conductivity of electrons ($K_{e}$) is proportional to the electrical conductivity according to the Wiedemann–Franz law (Equation (5)).
(5)$$K_{e}=\frac{\pi^{2}}{3}{\left(\frac{K_{B}}{e}\right)}^{2}\sigma T$$
where $K_{B}$, e, σ, and T are the Boltzmann constant, electron charge, electrical conductivity, and absolute temperature, respectively.

The electrical conductivity can be expressed by Equation (6) [37].
(6)$$\sigma =\frac{1}{aW^{3}+bW^{2}+cW+d}$$
where W is the thickness of the ICs, and a, b, c, and d are constants. According to Equations (5) and (6), the thermal conductivity is significantly influenced by the IC thickness. Thus, the Al–Cu composite sintered in a low-temperature range can have a thinner IC layer at the Al–Cu interface and exhibit higher thermal conductivity despite possessing a lower relative density. In the lower-temperature range, Al–Cu (380 °C) showed a high thermal conductivity of 150 W∙m^−1^∙K^−1^. The increase in thermal conductivity from 380 to 420 °C might be caused by grain growth. Therefore, the result that Al–Cu (380 °C) had a thermal conductivity comparable to the highest value among the Al–Cu composites prepared in this study suggests that an Al/Cu interface with controlled IC for high thermal conduction can be prepared at 380 °C.

The thermal conductivity of the Al–Cu composite is significantly affected by the IC layer thickness due to the numerous Al/Cu interfaces. Thus, it is possible to control the thermal conductivity by controlling the IC layer thickness. The IC layer thickness is determined by the interdiffusion mechanism, which has a strong dependence on the process temperature. From this study, we infer that low temperature sintering around 400 °C is suitable for preparing Al–Cu composites with a thin IC layer, and for effectively preventing the deterioration of the thermal conductivity.

## 4. Conclusions

We systemically investigated the interdiffusion mechanism at the Al/Cu interface and the effect of ICs on the mechanical and thermal properties of Al-50vol.%Cu composites prepared by solid-state sintering, with the process temperature ranging from 340 to 500 °C. We found that initially, a thin interdiffusion layer comprising ICs such as CuAl_2_ and Cu_9_Al_4_ was formed at the Al/Cu interface during the sintering of the Al–Cu composites. The thickness of the interdiffusion layer drastically increased at a sintering temperature of 400 °C, where the growth rate of Cu_9_Al_4_ might be higher than that of CuAl_2_ according to interdiffusion layer fractions from the Arrhenius equation. Al–Cu (380 °C) with a thin IC layer of 6–7 µm thickness exhibited a high Vickers hardness of 80 HV and thermal conductivity of 150 W·m^−1^·K^−1^ due to the controlled thin interdiffusion layer. We determined that low temperature sintering under 400 °C is the suitable process temperature for synthesising Al–Cu composites by powder metallurgy with remarkable mechanical and thermal properties and controlled growth of ICs at the Al/Cu interfaces. As interdiffusion can vary based on the particle size and the process used for preparing the mixing powder, this study suggests an appropriate process temperature range for the preparation of Al–Cu composites from ball-milled Al–Cu powder.

## Figures and Tables

**Figure 1 materials-14-04307-f001:**
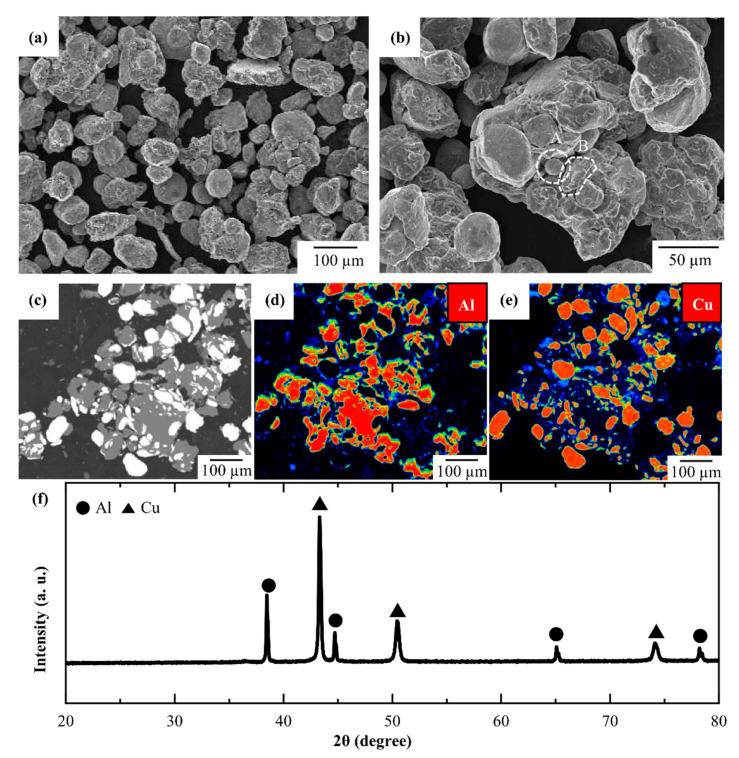
FE-SEM image of (**a**) Al–Cu powder prepared by ball milling and (**b**) high-magnification image of (**a**) with particles A and B labelled by dashed lines; (**c**) back scattered compositional image of Al–Cu powder; (**d**,**e**) FE-EPMA elemental mapping images for elements Al (**d**) and Cu (**e**); (**f**) XRD profile of Al–Cu powder.

**Figure 2 materials-14-04307-f002:**
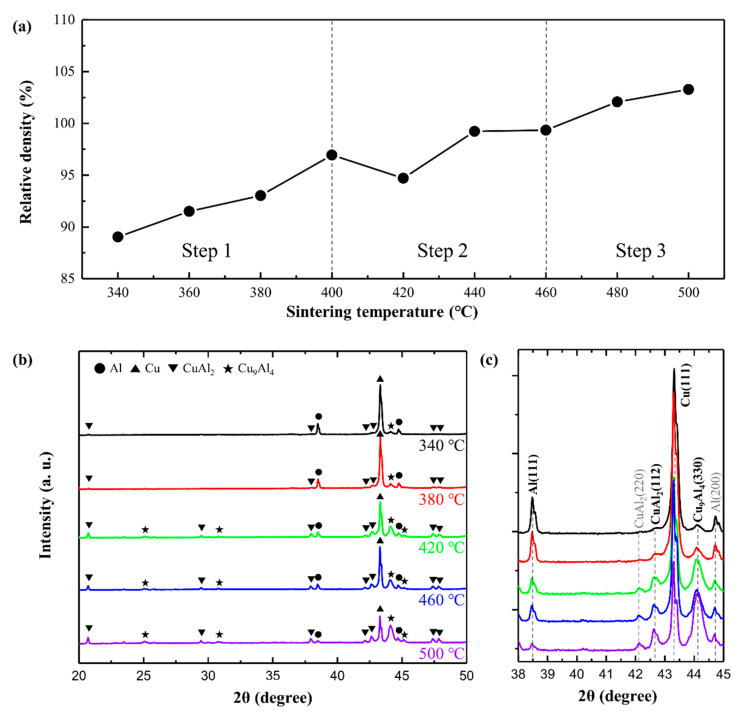
(**a**) Relative density of Al-50vol.%Cu composites as a function of sintering temperature (grey line indicates the expected density of the Al-50vol.%Cu composite calculated by the rule of mixtures); XRD profiles of Al-50vol.%Cu composites sintered at (**b**) 340, 380, 420, 460, and 500 °C in a wide 2θ scanning range of 35–55° and (**c**) 340, 380, and 420 °C in a narrow 2θ scanning range of 38–45°.

**Figure 3 materials-14-04307-f003:**
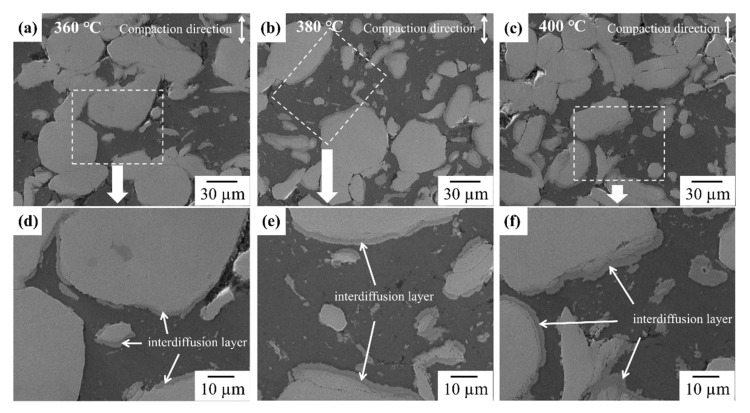
FE-SEM image of Al-50vol%Cu sintered at (**a**,**d**) 360 °C, (**b**,**e**) 380 °C, and (**c**,**f**) 400 °C.

**Figure 4 materials-14-04307-f004:**
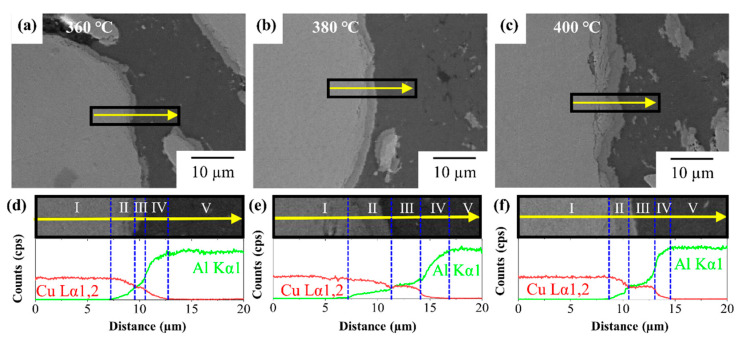
FE-SEM images and EDS line scan result with elements Al and Cu (green line: Al Kα 1, red line: Cu Lα 1,2) in cross-section of Al-50vol% Cu composites sintered at (**a**,**d**) 360 °C, (**b**,**e**) 380 °C, and (**c**,**f**) 400 °C.

**Figure 5 materials-14-04307-f005:**
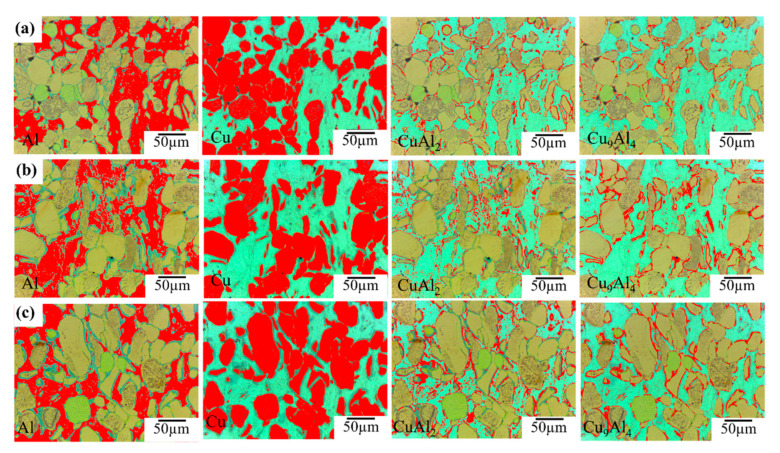
ImageJ analysis result with optical images: area of Cu, Al, CuAl_2_ and Cu_9_Al_4_ indicated by red colour in Al–50vol%Cu composites sintered at (**a**) 360 °C, (**b**) 380 °C, and (**c**) 380 °C.

**Figure 6 materials-14-04307-f006:**
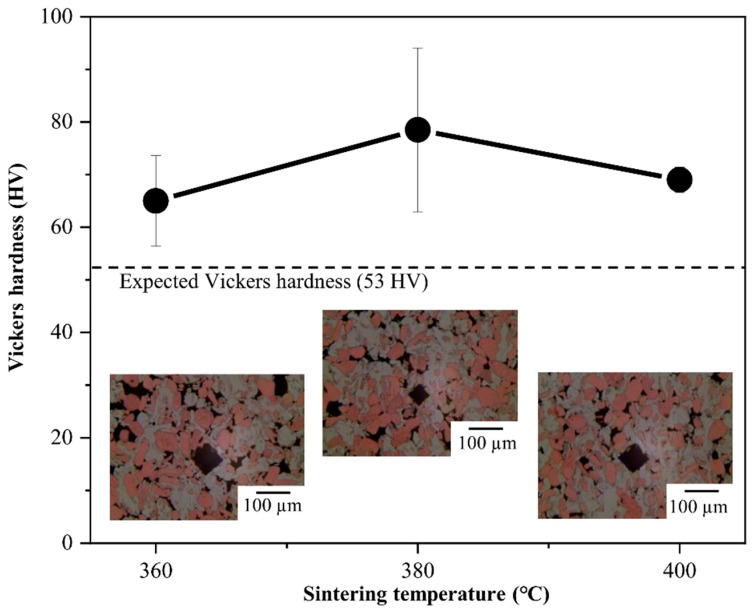
Vickers hardness with error bars of Al-50vol.%Cu composites as a function of sintering temperatures of 360, 380, and 400 °C; indentation optical microstructures of Al–Cu (360 °C), Al–Cu (380 °C), and Al–Cu (400 °C).

**Figure 7 materials-14-04307-f007:**
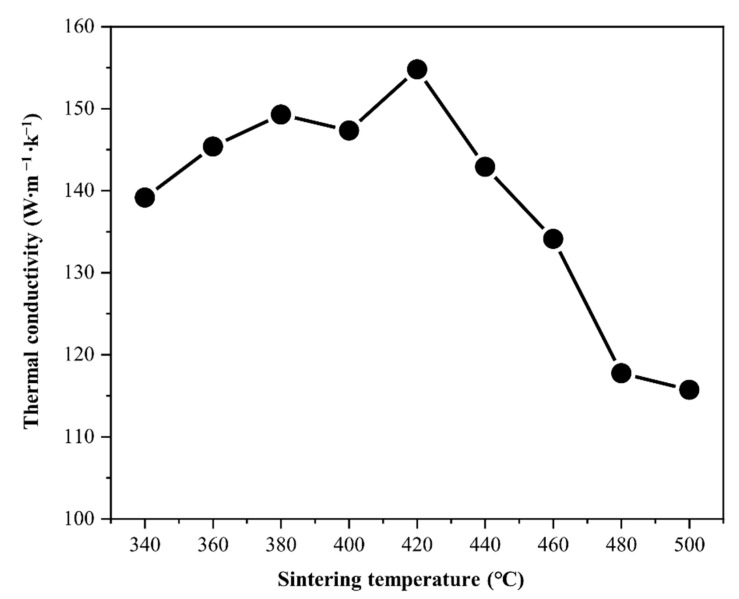
Thermal conductivity as a function of sintering temperature of the Al-50vol.%Cu composites sintered at 340, 360, 380, 400, 420, 440, 460, 480, and 500 °C.

**Table 1 materials-14-04307-t001:** Interdiffusion layer thickness fraction and area fraction of Al, CuAl2, Cu9Al4 and Cu.

Sintering Temperature (°C)	Thickness Fraction (%)			Area Fraction (%)			
	II	III	IV	Al	CuAl_2_	Cu_9_Al_4_	Cu
360	38	16	46	51	10	5	34
380	18	42	40	33	11	11	45
400	50	25	25	28	13	9	50

**Table 2 materials-14-04307-t002:** Density, Vickers hardness and thermal conductivity of Al–Cu composites.

Property	Sintering Temperature (°C)								
	340	360	380	400	420	440	460	480	500
Density (g∙cm^−3^) ± 0.0	5.2	5.3	5.4	5.7	5.5	5.8	5.8	6.0	6.0
Hardness (HV) ± 15.6	-	65.0	78.9	69.8	-	-	-	-	-
Thermal conductivity (W∙m^−1^∙K^−1^) ± 0.5	139.2	145.4	149.3	147.3	154.8	142.9	134.1	117.8	115.7

## Data Availability

Data sharing is not applicable to this article.

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
