# Peer review of "Interdiffusion and Intermetallic Compounds at Al/Cu Interfaces in Al-50vol.%Cu Composite Prepared by Solid-State Sintering"

_materials, 2021, doi:10.3390/ma14154307_

Round 1

Author Response

Thank you very much for taking your precious time to review our manuscript and giving us positive comments and suggestions. We tried to complement and revise the manuscript following your reviews as much as possible. The Responses to your reviews are prepared below. Please find the revised part highlighted with yellow color in the manuscript.

Reviewer 2 Report

The author has presented a comprehensive study on Al/Cu composite from powder to bulk. However, there are some issues need to be addressed in order to be published in the journal.

  1. in line 129 on page 4, the author claims the particle size exceed 100 um which is larger than the raw powder. The mixture of the Al and Cu particle is through ball milling at 300 rpm. Ball milling process is usually good at breaking down the particle size of the materials and making homogeneity of the particle size afterwards. Ball milling is not usually enlarging the particle size.
  2. In line 138, page 4, the cross-section of the ball-milled Al-Cu powder. Cross-section usually is used for bulk sample in SEM image. in figure 1 e and d, element mapping images of Al and Cu in red color. But no explanation for the appearance of blue colour.
  3. in line 143, the discussion on XRD on the mixture of Al and Cu does not need to explain and to claim the composite formation by ball milling. Because the powder mixture is very simple as no chemical reaction involved. XRD pattens showing pure Al and Cu is very obvious.
  4. in figure 2a, there is relative density on axis and expected relative density from rule of mixture. But this will make reader sometime confuse. The relative density is usually comparing the theoretical density not the raw material density calculated from the rule of mixture. When talk about relative density more than 100% and linked to porosity was not very accurate. Author should consider a simple and effective way to explain the density changing after sintering.
  5. in line 172 to 173 on page 6, I am not sure why author claim the Al is mainly consumed to form ICs as the ICs are made from both Al and Cu.
  6. in line 225-226, authors should explain where the six times coming from.
  7. For Vicker hardness test, the author should do multiple test on samples. It’s worth to compare the hardness of the different regions of the sample. For figure 6a, why is the error bar is much higher for sample sintered 380.

Author Response

(The authors gave the same response as above.)

Round 2

Reviewer 1 Report

The authors have revised the manuscript according to my comments. I  recommend to publish the current version of manuscript. 

Author Response

Thank you very much for taking your precious time to review our manuscript and giving us comments. We tried to complete and revise the manuscript following your reviews as much as possible. The Responses to your comments/reviews are prepared in the attached file.

Reviewer 2 Report

The authors have revised the manuscript according to my comments.  I am pleased with their reply and modifications in the manuscript and therefore I would recommended to published the current version in the journal. 

Author Response

(The authors gave the same response as above.)
